# Zero Endemic Cases of Wildlife Rabies (Classical Rabies Virus, RABV) in the European Union by 2020: An Achievable Goal

**DOI:** 10.3390/tropicalmed4040124

**Published:** 2019-09-30

**Authors:** Emmanuelle Robardet, Dean Bosnjak, Lena Englund, Panayiotis Demetriou, Pedro Rosado Martín, Florence Cliquet

**Affiliations:** 1French Agency for Food, Environmental and Occupational Health & Safety (ANSES), Nancy Laboratory for Rabies and Wildlife, European Union Reference Laboratory for Rabies, European Union Reference Laboratory for Rabies Serology, OIE Reference Laboratory for Rabies, WHO Collaborating Centre for Research and Management in Zoonoses Control, Technopôle agricole et vétérinaire de Pixérécourt, CS 40009, 54220 Malzéville, France; florence.cliquet@anses.fr; 2European Commission-Directorate-General for Health and Food Safety, B-1049 Brussels, Belgium; Dean.BOSNJAK@ec.europa.eu (D.B.); Lena.ENGLUND@ec.europa.eu (L.E.); Panayiotis.DEMETRIOU@ec.europa.eu (P.D.); Pedro.ROSADO-MARTIN@ec.europa.eu (P.R.M.)

**Keywords:** rabies, oral vaccination, red fox, vaccine, vaccination monitoring, rabies surveillance, Europe, European Union, elimination

## Abstract

The elimination of rabies transmitted by Classical Rabies Virus (RABV) in the European Union (EU) is now in sight. Scientific advances have made it possible to develop oral vaccination for wildlife by incorporating rabies vaccines in baits for foxes. At the start of the 1980s, aerial distribution of vaccine baits was tested and found to be a promising tool. The EU identified rabies elimination as a priority, and provided considerable financial and technical resources to the infected EU Member States, allowing regular and large-scale rabies eradication programs based on aerial vaccination. The EU also provides support to non-EU countries in its eastern and south eastern borders. The key elements of the rabies eradication programs are oral rabies vaccination (ORV), quality control of vaccines and control of their distribution, rabies surveillance and monitoring of the vaccination effectiveness. EU Member States and non-EU countries with EU funded eradication programs counted on the technical support of the rabies subgroup of the Task Force for monitoring disease eradication and of the EU Reference Laboratory (EURL) for rabies. In 2018, eight rabies cases induced by classical rabies virus RABV (six in wild animals and two in domestic animals) were detected in three EU Member States, representing a sharp decrease compared to the situation in 2010, where there were more than 1500 cases in nine EU Member States. The goal is to reach zero cases in wildlife and domestic animals in the EU by 2020, a target that now seems achievable.

## 1. Introduction

Classical terrestrial rabies caused by the rabies virus (RABV) has evolved for so long in the Old World that we are not certain of its exact origin [1]. The first written accounts date back to Antiquity, where rabies was prevalent and already reasonably well understood. The disease was described for the first time in the Eshuma code in Babylon, in the 23rd century BC (Before Christ), and then described by Hippocrates, Democritus, and Aristotle during Antiquity [2]. At that time, thanks to Celsius, the link between human rabies and animal bites by infected animals was established. RABV initially spread widely in Europe as a dog mediated disease. The number of deaths due to rabies for this period is unknown but many ancient texts mention tragic deaths among children, farmers, and hunters following dog or wolf attacks [2]. The disease started to decline gradually at the beginning of the 20th century, probably due to sanitary control measures involving restrictions on dog movements and the establishment of city pounds and shelters, but the exact reasons are not precisely documented [3].

Interestingly, rabies in Europe was often impacted by the geopolitical context. In the 1940s, during the Second World War, as a probable result of spill-over from domestic animals to wildlife, a new unexpected epizootic, circulating mainly in the red fox population (*Vulpes vulpes*), emerged in Eastern Europe [4]. The disease spread inexorably across Europe in all directions within a few decades with a speed of approximatively 15–60 km per year to reach France as the western-most country infected in 1968, and Italy in 1980 [5]. In the East, rabies spanned to all the countries that afterwards joined the European Union (EU). Large rivers, lakes, and high mountain ranges functioned as obstacles to the spread, but rivers were usually crossed where bridges were available, allowing a large spread of the disease [6]. Sweden, Ireland, and the United Kingdom have never had fox rabies, as well as some southern countries such as Portugal, Spain, Malta, and Cyprus [7]. Recently, the World Health Organization (WHO), the Food and Agriculture Organization of the United Nations (FAO), the World Organization for Animal Health (OIE), and the Global Alliance for Rabies Control (GARC) are joining forces to support countries as they seek to accelerate their actions towards the elimination of dog-mediated rabies by 2030 [8]. The efforts made in the European Union fulfils this Global Strategic Plan, targeting that no longer human death occurs due to this ancient and terrible disease.

## 2. Implementation of Oral Rabies Vaccination in Europe

The 20th century is probably one of the most remarkable periods with regard to rabies elimination in wildlife. To control the disease induced by the classical rabies virus (RABV), first attempts were made to reduce red fox populations, but they were proven ineffective to interrupt virus transmission [9,10]. In the 1970s, it was demonstrated experimentally that red foxes could be orally immunized against rabies using attenuated rabies viruses introduced in baits [11,12]. The concept of ORV was born and offered new perspectives of control [13]. The first ORV field trial was conducted in 1978 in Switzerland [14], and was followed by a succession of field trials in the 1980s in neighboring countries such as Belgium [15], France [6], and Germany [16]. In the meantime, control of rabies in red foxes in the EU became a challenge. More than 10,000 red fox cases were recorded annually in the early 1980s. The highest number of recorded fox case recorded was in 1990 with 12,425 cases (data source: Rabies Bulletin Europe). In the meantime, 2 humans, 525 dogs, 749 cats, and 743 cattle (data source: Rabies Bulletin Europe) were diagnosed rabid, showing the substantial risk that humans and domestic animals had to be affected during this epidemic. Oral immunization of wildlife through ORV started to be used on large scale in the EU thanks to EU funding since 1989 [17]. This tool was rapidly proven to be the only efficient technique in controlling the disease [18,19,20].

## 3. Role of the EU

The role of the EU has been threefold, comprising policy, funding, and technical support.

### 3.1. Rabies Eradication Policy

The European Commission (EC) is the executive of the European Union. In cooperation with the EU Member States, the EC identified rabies eradication as a priority. The final goal is the eradication of the disease from wildlife and domestic animals in the EU by 2020 [21].

The tools to implement the rabies eradication policy are the rabies eradication programs. The EC has fostered the implementation of programs based on large-scale aerial ORV campaigns in infected EU Member States [22]. They started in the late 80s of the past century, continuing until today. Also, the EC has promoted programs in countries bordering the EU such as the Russian Federation, Belarus, Ukraine, Moldova, Serbia, Montenegro, Albania, Bosnia and Herzegovina, Republic of North Macedonia, Kosovo (designation without prejudice to positions on status, and in line with UNSC 1244 and the ICJ Opinion on the Kosovo declaration of independence), and Turkey.

The programs consist of: ORV campaigns defining the vaccination areas, the vaccine(s) to be used and the method(s) of distribution, quality controls concerning the vaccine titre, maintenance of the cold chain when storing the vaccine and regular assessment of the vaccine bait distribution including coverage and vaccine bait density, rabies surveillance to assess the epidemiological trend of the disease, and monitoring to evaluate the effectiveness of the vaccination.

Annually, the competent authorities of the EU Member States may submit to the EC their draft programs. A group of external experts chaired by the EC assesses them. A key document for their preparation and assessment is the “Guidelines to design an EU co-financed programme on eradication and control of rabies in wildlife” [23]. Once it is deemed to be suitable from a technical point of view, the EC approves it and consequently allocates funds for its implementation. During the implementation, the EC follows closely the activities conducted by the competent authorities through interim and final reporting, and audits on the spot may take place. A summary of the audits performed by the EC can be found in the “Overview report Rabies Eradication in the EU” [24]. A list of technical indicators has been set up by the EC to support countries in their evaluation of the programs (Table 1). Following the implementation phase, there is a final technical and financial assessment of the annual reports submitted by EU Member States, to verify the right implementation and the adequate use of the EU funds.

### 3.2. Funding

Large-scale rabies eradication programs are possible thanks to the substantial EU funding support provided: As an example, in the period 2014–2019, € 131.4 million were allocated by the EU to eradicate rabies (14.4% of EU co-funding for animal disease eradication programs), being now the second most important disease after tuberculosis in terms of EU funding.

Within the EU, the measures that are eligible for EU funding are the purchase and distribution of vaccines, surveillance and monitoring tests, vaccine titration, collection and delivery of wild animals for testing, and awareness campaigns. These measures are subject to an EU support that ranges from 50% to 75% of the eligible costs.

In non-EU countries, the main measure eligible for EU support is the purchase and distribution of vaccines. This measure is subject to a 100% of EU support of the eligible costs.

### 3.3. Technical Support

To support EU Member States in the planning and in the implementation of the programs, the Task Force for monitoring disease eradication was established in 2000. Its main objectives are to progress towards animal disease eradication and to improve the cost-benefit ratio of animal disease eradication programs funded by the EU. Within the Task Force, the rabies subgroup started its activities in 2004. The rabies subgroup is composed of experts from EU Member States, the EU reference laboratory (EURL) for rabies Agence nationale de sécurité sanitaire, de l’alimentation, de l’environnement et du travail (ANSES), and the World Health Organization Collaborating Centre for Rabies Surveillance and Research Friedrich-Loeffler-Institute. Since 2010, its mandate has been expanded to non-EU countries. The rabies subgroup operates on the basis of field visits and provides a set of recommendations and conclusions addressed [25] to the competent authorities responsible for a program. They are issued following an in-depth assessment of the program, the manner it is implemented, and the rabies situation in a given country.

An important role is played by the EURL for rabies that was designated in 2008 with the aim to ensure high-quality and uniform testing in the EU, and to support EC activities in the area of laboratory analysis. The principal duties of the EURL are described in the Commission Regulation (EU) No 415/2013 of 6 May 2013, laying down additional responsibilities and tasks of the EURL for Rabies and amending Regulation (EC) No 737/2008 designating the EURL for Rabies. They include the organization of workshops, proficiency tests for the benefit of National Reference Laboratories [26,27,28], evaluation of critical reagents [29], evaluation of techniques [30,31,32] and trainings [33].

Finally, a key reference is the scientific report on the oral vaccination of foxes against rabies [34], covering the practical and organizational aspects to consider. The European Food Safety Authority (EFSA) updated this report with scientific panels in 2010 [35] and 2015 [36]. Such guidelines were drafted in line with WHO and OIE rabies recommendations [7,37,38,39,40,41].

## 4. Key Control Parameters of Vaccines and Vaccination

All rabies vaccine baits used in the EU for oral vaccination of wildlife need to fulfil the requirements of the European Pharmacopoeia monograph in terms of efficacy, safety, and stability [42], and need to have a marketing authorization in accordance with the EU legislation on veterinary medicinal products. The vaccine viral titers of all batches should be verified in qualified independent laboratories just before the ORV. The cold chain during transport and storage of vaccine baits has to be maintained as specified by the manufacturer. Storage under any other conditions prior to distribution in the field may have a serious negative impact on the vaccine titer. Such temperature data are usually controlled using temperature loggers (Figure 1).

The ORV campaigns are basically targeting red foxes and, to a lesser extent, raccoon dogs (*Nyctereutes procyonoides*). Raccoon dogs have indeed become the second most affected species in northeast Europe, particularly in the Baltic countries and in Poland. As the golden jackal (*Canis aureus*) could be a potential host [43], and following its recent geographical expansion [44], this species has been included since a few years of the monitoring of vaccination effectiveness. ORV campaigns are classically undertaken by aerial distribution of the vaccine baits twice a year, at a density around 20–25 baits/km^2^ [34,35,36]. Campaigns are organized in spring and autumn to match with the red fox biology [45] and optimal temperature conditions.

Evaluation of bait distribution is performed using GIS systems in airplanes with detection sensors for baits that can record every bait distributed during aerial distribution [46]. The data obtained during the distribution is assessed using GIS software, and bait distribution and density are calculated. This enables to establish precisely the bait coverage during the campaign in terms of both territory and density.

Rabies surveillance is another key aspect, and it is focused on testing indicator or suspect animals (indicator or suspect animals are animals that show clinical signs or abnormal behavior suggestive of rabies, animals found dead, road kills, and animal involved in human exposure). There is no sample size recommended, as all indicator or suspect animals should be analyzed. Follow up of the number of diagnosis tests performed and the number of rabies cases are paramount indicators of rabies surveillance system (Table 1). EU Member States are also engaged in the monitoring of vaccination effectiveness to verify the adequacy of the vaccination strategy. Vaccination monitoring is based on the assessment of the bait uptake through the detection of tetracycline in teeth, biomarker included in the vaccine bait, and seroconversion (herd immunity) in the target species. The objective is to sample and to analyze four target species animals/100 km^2^ of vaccinated area, starting the sampling one month after the ORV. Studies and field experience have demonstrated that vaccination campaigns should reach coverage of at least 70% of the susceptible population in order to break viral transmission [47]. Bait uptake and seroconversion evaluation assessment is necessary to possibly adapt the strategy. The sampling performed to evaluate vaccination effectiveness focuses on animals targeted by vaccination, and samples are collected from healthy hunted animals.

## 5. Outcome of the EU Rabies Eradication Programs

The implementation of these programs in the EU has produced a dramatic decrease in the number of rabies cases over the last 30 years (Figure 2). This has led to the elimination of rabies in the western and central part of the EU [5,16,48,49,50,51,52]. The disease is now confined to the eastern part of the EU, where it has been notified since 2016 only in four Member States: Lithuania, Poland, Hungary, and Romania.

As far as non-EU countries are concerned, the implementation of the programs resulted in the elimination of the disease in certain areas (Kaliningrad oblast of the Russian Federation or large areas of the Western Balkans), a decrease in the number of rabies re-incursions in the EU territory, and as an initial step for rabies elimination in these countries.

In 2018, while 21,707 animals were analyzed in the frame of passive surveillance, only eight endemic animal cases (six in wild animals and two in domestic animals) were detected in the EU territory: four cases in Poland, three cases in Romania, and one case in Lithuania (Figure 3) [53,54]. In 2018, rabies eradication programs [25] were still conducted in 12 Member States, including three Member States with endemic cases and nine countries bordering infected areas. The areas planned to be vaccinated in 2018 are in Figure 4. An area of approximately 832,991 km^2^ was vaccinated with 19,500,761 baits corresponding to a global bait density of 23 baits per km^2^ [54]. As a result of the monitoring, 72% of bait uptake and 49% of seroconversion was estimated on the EU territory [54]. The seroconversion rate should be considered with caution, as different types of tests with variable sensitivity/specificity are used depending on the country. Forty-seven vaccine batches were controlled before distribution, and all except one were found satisfactory. The last human endemic case in EU Member States was in Romania in 2012.

## 6. Conclusions

The 21st century will be the one of the elimination of wildlife rabies from the EU territory, which would be unprecedented achievement in the history of this animal disease. The long-term maintenance of the rabies-free status will be challenging, as many bordering non-EU countries are still infected [55]. Keeping an immune belt consisting of vaccination strips along the EU borders inside EU Member States and of vaccination strips in bordering areas inside the neighboring non-EU countries [21] should prevent reintroductions. Maintaining adequate rabies surveillance despite a context of disappearance of the virus on the EU territory will be the most challenging issue, although this component is crucial.

Finally, in the light of the EU experience, we would like to underline some key issues that are essential to eradicate the disease:Eradication programs should be conducted in large areas and maintained at least six years, and be stopped no earlier than two years after the last confirmed case;The vaccine to be used should be carefully selected and compliant with the legislation on veterinary medicines;The distribution of the vaccine should be mainly aerial and controlled on a regular basis to detect early areas where the bait density is not sufficient, in order to take immediate corrective actions;The surveillance and monitoring components of the program are critical to know whether rabies is being eradicated and whether the vaccination is being effective. Therefore, the collection of samples from an adequate number of animals is of foremost importance;Involvement of all stakeholders, including hunter associations and rural communities is necessary;International and cross-border cooperation in the planning and implementation of the eradication programs is needed to ensure a coordinated approach and the full vaccination of bordering areas.

## Figures and Tables

**Figure 1 tropicalmed-04-00124-f001:**
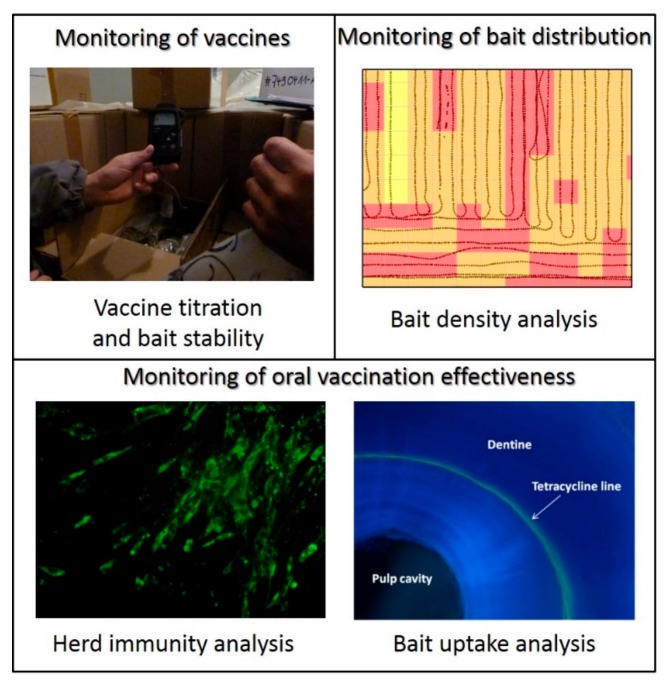
Quality control parameters used in the EU to evaluate the implementation and efficacy of oral vaccination programs.

**Figure 2 tropicalmed-04-00124-f002:**
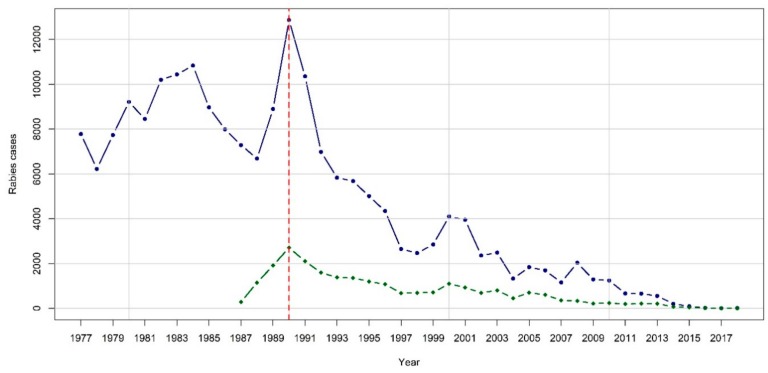
Evolution of fox cases in the 28 EU countries (data source: Rabies Bulletin Europe, Rabies Information System of the WHO Collaboration Centre for Rabies Surveillance and Research; http://www.who-rabies-bulletin.org/Queries/Default.Aspx). Red dashed line: 1989, start of European Commission co-funding and large scale ORV programs; Blue line: fox cases; Green line: domestic cases.

**Figure 3 tropicalmed-04-00124-f003:**
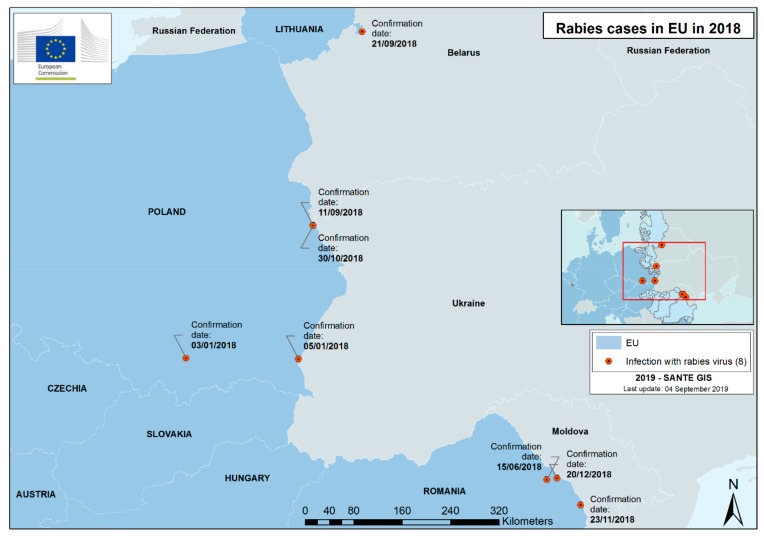
Rabies cases detected in the European Union in 2018.

**Figure 4 tropicalmed-04-00124-f004:**
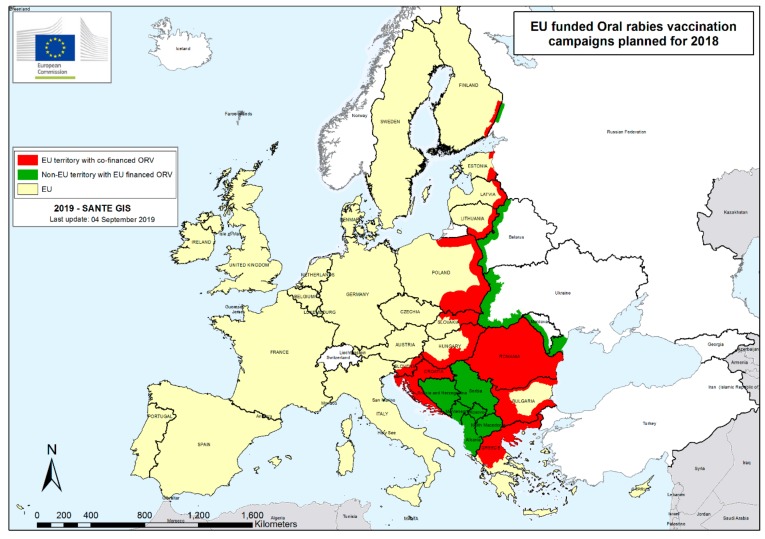
Oral rabies vaccination campaigns planned in 2018.

**Table 1 tropicalmed-04-00124-t001:** Indicators for the evaluation of rabies eradication, control and surveillance programs.

Activities Indicators
Number of vaccine batches controlled before distribution
Number of oral vaccination campaigns performed within the year
Total number of baits distributed per campaign
Density per campaign (number of baits per square kilometer distributed)
Area covered with oral rabies vaccination per campaign
Number of monitoring tests for vaccination effectiveness on target species per campaign
Number of surveillance tests performed (passive surveillance)
**Progress Indicators**
Number of rabies cases (excluding bat cases) compared to the previous year
Number of rabies cases in previously (last year) case-free areas compared to previous year
Percentage of seroconversion in target species (juveniles /adults) compared to previous year
Percentage of vaccine uptake in target species (juveniles/adults separately) compared to the previous year.

Source: https://ec.europa.eu/food/sites/food/files/animals/docs/diseases_sanco-12915-2012_en.pdf.

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
