# Peer review of "Zero Endemic Cases of Wildlife Rabies (Classical Rabies Virus, RABV) in the European Union by 2020: An Achievable Goal"

_tropicalmed, 2019, doi:10.3390/tropicalmed4040124_

Round 1

Reviewer 1 Report

This is a review paper describing the role played by the European Union (EU) in the success of fox rabies elimination across Europe. Given that we are now on the eve of the designated deadline set for this elimination agenda, the review is both timely and addresses a critically important issue involving decades of effort and significant levels of funding and co-funding by the EU. The overall success of this program is impressive and serves as an important proof of concept that oral rabies vaccination strategies can successfully eliminate RABV from wildlife across very broad regions. The effort also serves as a prime example of highly productive intersectoral and multilateral cooperation and collaboration to combat a wildlife disease with cutting-edge science, potentially serving as a model for the control of other wildlife diseases. The paper is concise yet well written and I have only minor suggestions for improving clarity or description of certain elements.  

L30-31 – I suggest the authors reference doi:10.1371/journal.pntd.0005266,

 doi:10.1038/nrmicro.2018.11, or some equivalent works regarding hypotheses on the origins of RABV.

L61 – perhaps the authors could elaborate briefly on the public/animal health impact of fox rabies at this time (where >10,000 cases were reported annually).

L63 – suggest replacing ‘from’ with ‘beginning in’ or ‘since’

L67 – while I recognize that this may be terminology used by the EC, eradication seems an inappropriate word in the context here and throughout the paper. While RABV will very likely be eliminated from the EU imminently, it seems premature to suggest it will be eradicated.

L73 – the publication by Stohr and Meslin Vet Record 1996 139:32-35 may be an appropriate reference for consideration in describing early ORV program development across Europe.

L75 – should the “i” be replaced with “&” in describing Bosnia and Herzegovina

L78-85 – I suggest that this section be formatted as part the preceding paragraph. It reads a bit awkwardly as formatted here.

L103-104 – should the commas be replaced with periods-  i.e., 131.4 Million and 14.4%?

L112 – in L66 this is termed “technical support” – I recommend consistency with this sub-heading.

L122-125 – it appears this section can be joined as part of the preceding paragraph and need not be a stand-alone paragraph

L126 – please introduce this abbreviation where it is spelled out in L117/118

L133-135 – surely the WHO (1992, 2005, 2013, 2018) and OIE Terrestrial Manual (2018) guiding documents on rabies are worth mention here as well – if nothing more than to illustrate the EU/EC documents build upon the principles outlined by WHO/OIE?

L148 – the scientific (Latin) name of the red fox has already been introduced on L44

L149 – are raccoon dogs targeted throughout the EU, or perhaps the authors can clarify geospatially where raccoon dogs have been of epizootiological importance.  

L157-159 – please clarify the relative roles of public health versus enhanced surveillance for this purpose. Please also clarify the spatiotemporal sampling guidelines if the design is not entirely opportunistic.

L157-166 – please also describe what is done downstream with all this information.

L161 – please clarify that bait uptake is measured by biomarkers. This is not explicitly mentioned.

L162 – please clarify that the seroconversion is measured post-ORV. Of course this should be implied, but I would argue that it is important to specify even the window post-ORV recommended for this monitoring activity in the EU. Is the activity strictly opportunistic by design or there is some structure to the sampling worth mentioning here?

L183-184 – please clarify the denominator of cases tested to clarify that adequate surveillance efforts are still in place.

L188-189 – please provide a salient description of published studies on broad-scale program evaluation? For example, http://dx.doi.org/10.1098/rstb.2012.0142

Figure 1 – please label the four panels of this figure and explain what the reader should be interpreting from these images.

Figure 3 – please increase the font of the text labels. Please include standard items such as a compass and scale bar. It would be great if the negative cases could also be plotted (within the currently shown geographic area) for visual comparison.

Figure 4 – Please include standard items such as a compass and scale bar. It would be relevant to clarify what is meant by the green shading and label “buffer zones in non-EU countries”. Are these areas also conducting ORV?

Author Response

Thank you for accepting the review and for suggestions of improvement given opportunity to improve the manuscript.

L30-31 – I suggest the authors reference doi:10.1371/journal.pntd.0005266,

 doi:10.1038/nrmicro.2018.11, or some equivalent works regarding hypotheses on the origins of RABV.

The paper has been modified accordingly.

L61 – perhaps the authors could elaborate briefly on the public/animal health impact of fox rabies at this time (where >10,000 cases were reported annually).

Data on impact of animals and humans (rabies cases) has been indicated.

L63 – suggest replacing ‘from’ with ‘beginning in’ or ‘since’

The paper has been modified accordingly.

L67 – while I recognize that this may be terminology used by the EC, eradication seems an inappropriate word in the context here and throughout the paper. While RABV will very likely be eliminated from the EU imminently, it seems premature to suggest it will be eradicated.

The term “eradication” has been replaced by “elimination” all along the manuscript excepted when “Eradication programmes are mentioned” as it is indeed the terminology used by EC for EC founded programmes. European Commission wish to conserve terminology used in documents, as we do not have elimination programmes but eradication programmes.

L73 – the publication by Stohr and Meslin Vet Record 1996 139:32-35 may be an appropriate reference for consideration in describing early ORV program development across Europe.

The paper has been modified accordingly.

L75 – should the “i” be replaced with “&” in describing Bosnia and Herzegovina

Bosnia i Herzegovina is the correct spelling.

L78-85 – I suggest that this section be formatted as part the preceding paragraph. It reads a bit awkwardly as formatted here.

The paper has been modified accordingly.

L103-104 – should the commas be replaced with periods-  i.e., 131.4 Million and 14.4%?

The paper has been modified accordingly.

L112 – in L66 this is termed “technical support” – I recommend consistency with this sub-heading.

The paper has been modified accordingly.

L122-125 – it appears this section can be joined as part of the preceding paragraph and need not be a stand-alone paragraph

The paper has been modified accordingly.

L126 – please introduce this abbreviation where it is spelled out in L117/118

The paper has been modified accordingly.

L133-135 – surely the WHO (1992, 2005, 2013, 2018) and OIE Terrestrial Manual (2018) guiding documents on rabies are worth mention here as well – if nothing more than to illustrate the EU/EC documents build upon the principles outlined by WHO/OIE?

The paper has been amended accordingly.

L148 – the scientific (Latin) name of the red fox has already been introduced on L44

The paper has been modified accordingly.

L149 – are raccoon dogs targeted throughout the EU, or perhaps the authors can clarify geospatially where raccoon dogs have been of epizootiological importance.

Information where raccon dogs have epizootiological importance has been added in the manuscript.

L157-159 – please clarify the relative roles of public health versus enhanced surveillance for this purpose. Please also clarify the spatiotemporal sampling guidelines if the design is not entirely opportunistic.

The paper has been amended accordingly.

L157-166 – please also describe what is done downstream with all this information.

The information has been added in the manuscript.

L161 – please clarify that bait uptake is measured by biomarkers. This is not explicitly mentioned.

The paper has been amended accordingly.

L162 – please clarify that the seroconversion is measured post-ORV. Of course this should be implied, but I would argue that it is important to specify even the window post-ORV recommended for this monitoring activity in the EU. Is the activity strictly opportunistic by design or there is some structure to the sampling worth mentioning here?

The information has been added in the manuscript.

L183-184 – please clarify the denominator of cases tested to clarify that adequate surveillance efforts are still in place.

The information has been added in the manuscript.

L188-189 – please provide a salient description of published studies on broad-scale program evaluation? For example, http://dx.doi.org/10.1098/rstb.2012.0142

The reference has been added in the manuscript.

Figure 1 – please label the four panels of this figure and explain what the reader should be interpreting from these images.

The figure 1 has been updated accordingly.

Figure 3 – please increase the font of the text labels. Please include standard items such as a compass and scale bar. It would be great if the negative cases could also be plotted (within the currently shown geographic area) for visual comparison.

The figure 3 has been updated accordingly. Negative cases localization are unfortunately not available.

Figure 4 – Please include standard items such as a compass and scale bar. It would be relevant to clarify what is meant by the green shading and label “buffer zones in non-EU countries”. Are these areas also conducting ORV?

The figure 4 has been updated accordingly.

Reviewer 2 Report

The manuscript is very well written and clear, and it provides a very interesting summary of the history and actions taken for rabies eradication in European Union.

I consider the manuscript have the potential to be published but I have some major comments and some minor ones, for which I recommend the paper for publication after major revision.

Major comments.

-The paper described very well the eradication policy in the European Union and the steps that are currently allowing the eradication of classical rabies (RABV) in the territory of the EU. The paper described how the progress of the eradication policy was monitored through ACTIVITIES INDICATORS and PROGRESS INDICATORS (Table 1). However, the "section 4. Outcome of the EU rabies eradication programmes" presents only data about "Number of rabies cases (excluding bat cases) compared to the previous year". I think more work is needed to expand this section, in order to present results on the different activities and progress indicators.

- In general the paper should state more clearly that the topic of the paper, and the objective of the eradication project is classical rabies virus, genotype-1 (RABV), and that it does not refer to other genotypes of Rabies lyssavirus.   

Minor comments. 

-I would suggest, in the introductory part, to make reference to the "Global framework for the elimination of dog-mediated human rabies" among the World Organisation for Animal Health (OIE), the World Health Organization (WHO), the Food and Agriculture Organization of the United Nations (FAO) and with the support of the Global Alliance for Rabies Control (GARC).

-Line 2-3: should be indicated that the paper refers to classical (RABV) only.

-Line 12-13: vaccination baits are not specific for foxes, they can be eaten by each carnivore (and not only) species entering in contact with them.

-Line 23: 12 cases according to rabies bulletin Europe in the 4 affected countries. Lithuania, Poland, Hungary and Romania. Also, please specify that the number refers to classical (RABV) rabies cases.

-Line 23: how many cases in domestic animals?

- Line 60: the authors sometimes speak about control of rabies in fox, some others of control of rabies in wildlife. Even if most of the cases are reported in foxes, I think there is a need to be consistent, speaking about control of classical rabies in wildlife through all the manuscript.

-Line 148: it is technically not possible to target vaccination on a specific wildlife species. Baits are distributed and then several species can enter in contact with the vaccine.

-Line 148-149: what about Canis aureus that demonstrate also to be a competent host for rabies and it is expanding its population in Europe?

-Line 171: please specify if you referred to rabies cases in domestic or wildlife. Apart from rabies cases in wildlife, Hungary reported 2 cases in goats in 2017 , and Romania reported several cases in dogs and cattle in the period 2016 - 2018 to the OIE (WAHIS system).

-Line 183-184: different figures are reported in the rabies bulletin Europe. Please check.

Author Response

Thank you for accepting the review and for suggestions of improvement given opportunity to improve the manuscript.

Information regarding activities and progress indictors (for 2018) has been added in the text.

The title of the document has been modified to clarify that only classical rabies virus RABV is considered in EC eradication programmes and in this document. The document has been revised and modified in order to avoid as much as possible misunderstanding.

Minor comments. 

-I would suggest, in the introductory part, to make reference to the "Global framework for the elimination of dog-mediated human rabies" among the World Organisation for Animal Health (OIE), the World Health Organization (WHO), the Food and Agriculture Organization of the United Nations (FAO) and with the support of the Global Alliance for Rabies Control (GARC).

-Line 2-3: should be indicated that the paper refers to classical (RABV) only.

The paper has been amended accordingly.

-Line 12-13: vaccination baits are not specific for foxes, they can be eaten by each carnivore (and not only) species entering in contact with them.

The vaccine baits are indeed not specific to red foxes as other animals can consumed the bait. Since the fox is the main rabies host in Europe, vaccine baits have however been designed, evaluated and validated to be used on red fox and the vaccination strategy has been reflected to reach foxes which are considered to be the main target species of vaccination campaigns.This is how we considered the term “target species”, as the intent, the aim of the campaign, but not the result.

-Line 23: 12 cases according to rabies bulletin Europe in the 4 affected countries. Lithuania, Poland, Hungary and Romania. Also, please specify that the number refers to classical (RABV) rabies cases.

Hungarian case occurred in 2017.

For 2018, 9 cases are recorded in the Rabies Bulletin Europe (4 Romanie, 4 Poland, 1 Lithuania).

According to EU NRLs annual review and ADNS system of the European Union, 8 cases are recorded (only 3 cases were recorded in Romania). This has been cheeked with the national Reference Laboratory for Rabies in Romania who confirmed that only 3 cases occurred.

-Line 23: how many cases in domestic animals?

Two. The information has been added in the manuscript.

- Line 60: the authors sometimes speak about control of rabies in fox, some others of control of rabies in wildlife. Even if most of the cases are reported in foxes, I think there is a need to be consistent, speaking about control of classical rabies in wildlife through all the manuscript.

Monitoring evaluation is performed on red foxes, raccoon dogs in Northeast of Europe and golden jackals only (only recently for golden jackals). This information has been added in the manuscript.

-Line 148: it is technically not possible to target vaccination on a specific wildlife species. Baits are distributed and then several species can enter in contact with the vaccine.

The vaccine baits are indeed not specific to red foxes as other animals can consumed the bait. Since the fox is the main rabies host in Europe, vaccine baits have however been designed, evaluated and validated to be used on red fox and the vaccination strategy has been reflected to reach foxes which are considered to be the main target species of vaccination campaigns.This is how we considered the term “target species”, as the intent, the aim of the campaign but not the result.

-Line 148-149: what about Canis aureus that demonstrate also to be a competent host for rabies and it is expanding its population in Europe?

This information and references have been added in the manuscript.

-Line 171: please specify if you referred to rabies cases in domestic or wildlife. Apart from rabies cases in wildlife, Hungary reported 2 cases in goats in 2017 , and Romania reported several cases in dogs and cattle in the period 2016 - 2018 to the OIE (WAHIS system).

-Line 183-184: different figures are reported in the rabies bulletin Europe. Please check.

Number of domestic and wildlife cases has been specified. WE agree Hungarian case occurred in 2017. The manuscript report on data for 2018 only. For 2018, 9 cases are recorded in the Rabies Bulletin Europe (4 Romanie, 4 Poland, 1 Lithuania).

According to EU NRLs annual review and ADNS system of the European Union, 8 cases are recorded (only 3 cases were recorded in Romania). This has been cheeked with the national Reference Laboratory for Rabies in Romania who confirmed that only 3 cases occurred.

Round 2

Reviewer 2 Report

I would like to thank the authors for accepting my suggestion and applied the requested modifications to the text.

I consider in the presence the paper is suitable for publication